# Nanoparticles-Based Oligonucleotides Delivery in Cancer: Role of Zebrafish as Animal Model

**DOI:** 10.3390/pharmaceutics13081106

**Published:** 2021-07-21

**Authors:** Sara Bozzer, Michele Dal Bo, Giuseppe Toffoli, Paolo Macor, Sara Capolla

**Affiliations:** 1Department of Life Sciences, University of Trieste, 34127 Trieste, Italy; sara.bozzer@phd.units.it; 2Experimental and Clinical Pharmacology Unit, Centro di Riferimento Oncologico di Aviano (CRO), IRCCS, 33081 Aviano, Italy; mdalbo@cro.it (M.D.B.); gtoffoli@cro.it (G.T.); sara.capolla@cro.it (S.C.)

**Keywords:** oligonucleotides, nanoparticles, zebrafish

## Abstract

Oligonucleotide (ON) therapeutics are molecular target agents composed of chemically synthesized DNA or RNA molecules capable of inhibiting gene expression or protein function. How ON therapeutics can efficiently reach the inside of target cells remains a problem still to be solved in the majority of potential clinical applications. The chemical structure of ON compounds could affect their capability to pass through the plasma membrane. Other key factors are nuclease degradation in the extracellular space, renal clearance, reticulo-endothelial system, and at the target cell level, the endolysosomal system and the possible export via exocytosis. Several delivery platforms have been proposed to overcome these limits including the use of lipidic, polymeric, and inorganic nanoparticles, or hybrids between them. The possibility of evaluating the efficacy of the proposed therapeutic strategies in useful in vivo models is still a pivotal need, and the employment of zebrafish (ZF) models could expand the range of possibilities. In this review, we briefly describe the main ON therapeutics proposed for anticancer treatment, and the different strategies employed for their delivery to cancer cells. The principal features of ZF models and the pros and cons of their employment in the development of ON-based therapeutic strategies are also discussed.

## 1. Introduction

In recent years, molecular target agents, constituted by chemically synthesized antisense oligonucleotides (ASOs), small interfering RNA (siRNA), microRNA (miRNA), aptamers, and decoys, have been proposed as interesting therapeutic compounds for anticancer treatments [1]. They are based on chemically synthesized oligonucleotides (ONs) with a single- or double-stranded deoxyribonucleic acid (DNA) or ribonucleic acid (RNA) chain with a specific molecular target. This target can be represented by a specific gene, protein, or a class/family of genes or proteins [2]. In this context, the various molecular profiling strategies that have recently been introduced in the cancer field of studies have dramatically increased the range of molecular targets [3]. The capability to be highly specific for a particular molecular target can be of great interest in the case of anticancer therapies in which other conventional or innovative drugs are inefficacious. In the last two decades, diverse advances have been made in the potential application fields of ON technologies with both diagnostic or therapeutic purposes, particularly due to the possibility of acting at different stages of tumor pathogenesis and progression for the different mechanisms of action [4,5,6]. In particular, ASO, siRNA, and miRNA can act at transcription level by targeting specific messenger RNAs (mRNAs), aptamers are capable of directly inhibiting the activity of proteins whereas decoys are capable of targeting specific DNA coding transcription factors (Table 1) [4,5,6].

Although a significant number of phase III clinical trials based on ONs have already been proposed, aspects related to their efficacy still limit the introduction of ON-based compounds in clinical practice for anticancer treatment [2]. In particular, further studies are needed to understand better drug delivery strategies for each proposed ON-based compound and to develop in vivo models in which the efficacy and the toxicity of these approaches can be evaluated. In this context, the employment of zebrafish (ZF) models could expand the range of possibilities to understand the efficacy of the investigated therapeutic strategies [7].

In the present review, we summarize the main ON therapeutics that have been proposed for anticancer treatments. We also describe the different structures employed for the delivery of these ON compounds to the cancer cells as well as the principal features of ZF models and the pros and cons of their employment as in vivo models in the development of ON-based therapeutic strategies.

## 2. ON Therapeutics

### 2.1. Antisense Oligonucleotides (ASOs)

ASOs are synthetic, single-stranded nucleic acid polymers of various chemistries with a size of about 18–30 nucleotides. Depending on their mechanisms of action, they can be divided into two different groups, that is, RNase competent and steric block.

In the case of RNase competent, the degradation of RNA is catalyzed by the RNase H enzyme RNASEH1 in presence of the RNA-DNA duplex formed by the binding of the DNA-based ONs with their specific mRNA transcripts. Thus, the target gene expression is silenced (Figure 1A) [8,9]. A central DNA-based “gap” surrounded by flanking regions, constituted by chemically modified RNA promoting target binding, constitutes the recently proposed RNase H-competent ASOs [10]. Being RNASEH1 active in both the cytoplasm and the nucleus, this technology enables the targeting of nuclear transcripts—in particular for immature pre-mRNAs and long non-coding RNAs-while for other technologies, such as siRNA, the silencing in the nucleus context is more difficult [11,12].

Steric block ONs represent the second group of ASOs. They can interfere with either transcript RNA-RNA, RNA-protein, or both, interactions by masking specific sequences within a target transcript. The modulation of alternative splicing to selectively choose a specific transcript by excluding or retaining specific exon(s) is so far one of the most employed applications of this technology. This results in the alteration of splicing because of the masking of a splicing signal inhibiting the spliceosome function (Figure 1B).

The described approach has been suggested for rescuing of the production of a protein with a therapeutic functionality [13,14]. Moreover, the same approach has been also used to disrupt the translation of a specific target gene. Splicing modulation, achieved through this approach, can also make it possible to select the preferred isoform for a selected protein [15,16]. Steric block ONs can also be employed for the neutralization of translation inhibition interfering with upstream open reading frames that negatively regulate translation [17,18,19].

### 2.2. siRNA

It is possible to cause the inhibition of RNA using siRNA molecules. They are constituted by a duplex of two RNA molecules of 21 nucleotides where 19 complementary bases and two 3′ overhangs of 2 nucleotides are present [20]. The siRNA is the guide or antisense strand that is complementary to the target transcript, while the other strand is the passenger or sense strand. siRNA molecule activity is carried out guiding the Argonaute2 protein (AGO2) in the context of the RNA-induced silencing complex (RISC) to the complementary target transcripts. siRNA and target transcript complete complementarity determines the cleavage of the target opposite position of the guide strand catalyzed by AGO2, that results in gene silencing (Figure 2) [21,22,23].

Several approaches have been introduced to ameliorate the efficacy of siRNA including single stranded siRNAs, divalent siRNAs, Dicer substrate siRNAs, small internally segmented siRNAs, and self-delivering siRNAs [24,25,26,27,28,29].

### 2.3. miRNAs

The overexpression of miRNAs or their underexpression can be found in various tumor types and disease states. miRNA mimics and anti-miRNAs (antagomiRs) are two approaches developed in order to modulate a specific miRNA level (Figure 3). For the purpose of restoring the lowered expression of endogenous miRNA associated with tumor suppressor functions, miRNA mimics are synthetized as miRNA duplexes whose sequences are the same as the endogenous miRNA. Specific steric block ASOs, termed antagomiRs, are able to effectively silence miRNAs through direct binding within the RISC complex [30,31]. Alternatively, miRNAs’ regulatory functions can be inhibited by masking the respective target sequence on the mRNA transcript with the use of steric block ASO [31].

### 2.4. Aptamers

About 20–100 nucleotides folding into defined secondary structures comprise the aptamers, consisting of single-stranded nucleic acid molecules. Thanks to their three-dimensional structure, aptamers can bind with target proteins (Figure 4).

Differently from the ONs used for therapeutic purposes for which the therapeutic target can only be intracellular, aptamer therapeutics can be developed for intracellular, extracellular, or cell-surface targets. In particular, for these latter two classes of possible targets, there is no necessity to cross the cell membrane. Thus, in a similar way to monoclonal antibodies, aptamers can be used with therapeutic purposes in any disease for which extracellular blockade of protein–protein interactions is required, including anticancer treatments. In the context of anticancer treatments, aptamers have been proposed for their capability to reach the tumor site in difficult contexts, as in the case of solid tumors or metastasis. Aptamers have also been proposed as delivery tools to enhance anti-tumor immunity by the expression of immunogenic neoantigens [32,33,34]. The methodology to produce aptamers in vitro is called systematic evolution of ligands by exponential enrichment (SELEX) [35,36,37]. The SELEX methodology can be carried out through the use of libraries of chemically modified RNAs with nucleotide analogues, namely, 2′-fluoro or 2′-*O*-methyl recognized in the reverse transcriptase and T7 RNA polymerase steps. Alternatively, in order to increase the efficacy of aptamers as therapeutic compounds, chemical modifications can be introduced successively to the SELEX process [38]. Of note, in many cases, a developed aptamer selected to bind to a specific protein is also capable of inhibiting its function. A possible explanation for this is that the protein active sites show more exposed heteroatoms for hydrogen bonding and other interactions. An alternative possibility is that since aptamers can have a limited number of possible interactions with a protein target, they are more likely selected when they fit into a crevice on a protein, such as an active site, according to the so-called homing principle [35,36,37]. Thus, aptamers generally function as antagonists as they tend to inhibit protein–protein interactions, such as receptor–ligand interactions. However, in some cases, aptamers can act as agonists, as in the case of aptamers isolated against the extracellular domain of the protein human epidermal growth factor receptor 3 (HER3). On the other hand, it has been demonstrated that a DNA aptamer, isolated against an isoleucyl tRNA (RNA transfer) synthetase, is capable of enhancing editing activity. In order to increase target affinity and specificity, as well as the capability to show a long half-life when present in the biological target site, several SELEX methodologies have been proposed. This is particularly important for aptamers with antagonist functions, given that in this case the therapeutic effects can be present as long as the aptamer can physically dock with the target. Thus, the therapeutic effect can be more prolonged in the case of higher binding affinity [33].

### 2.5. Decoys

A double-stranded DNA or RNA structure characterizes decoys, which are able to inhibit the DNA transcription as they block the activity of double stranded-binding transcription factors (Figure 5). Specifically, they are capable of mimicking the specific region of a target transcription factor related to the binding site in the cis-regulatory promoter sequence of a target gene. Even so, the expression of the transcription factor itself cannot always be blocked by these molecules [39,40].

## 3. Platforms for the Delivery of Oligonucleotides (ONs)

ONs mainly act within the intracellular space. There are biological barriers that have to be overcome by ONs to reach the target when administered in vivo. In particular, in the case of systemic administration a limit may be represented by the tissue uptake that has to be maximized, whereas the exposure to other tissues not representing the target has to be avoided [41]. Moreover, the cellular uptake of ONs mainly occurs via different types of endocytosis through which ONs subsequently move into the endolysosomal system [42], and traffic into multiple membrane-bound intracellular compartments [43]. There are membrane barriers capable of separating the ONs accumulated by cells from the cytosol and the nucleus, thus determining both effective and non-effective pathways of cellular uptake. In particular, different endocytosis processes have been related in the uptake of ONs [44]. One of them is the coated pit pathway that utilizes adaptor proteins, a clathrin network and the guanosine triphosphate hydrolase (GTPase) dynamin to concentrate ligand-bound receptors at the cell surface and direct them into the cells [45]. Moreover, compact structures named caveolae, originating from membrane structures with a high quantity of cholesterol, sphingolipids, and caveolin, have been found to play a role in the internalization of several receptors and ligands [46]. The fluid phase endocytosis is exerted by the cells through the formation of tubular endosomes by the action of the clathrin-independent carrier/GPI-anchored protein-enriched early endosomal compartment (CLIC/GEEC) pathway. Moreover, large amounts of cellular fluid can be pinched off and engulfed by cells through an actinomyosin-driven process called macropinocytosis [47,48]. The trafficking through a complex network of endomembrane compartments is the second step of endocytosis. These compartments are represented by early and recycling endosomes, lysosomes, the Golgi apparatus, and the endoplasmic reticulum [49]. As mentioned above, the first step of internalization is made by early endosomes. Then, ONs can be shunted to lysosomes or recycled to the plasma membrane and the cell exterior. In many cases, the early endosome is where the dissociation between receptor and cargo happens [50]. Lysosomes are dense organelles that thanks to the action of hydrolases in a 4.5–5.5 pH range are the sites in which proteins, lipids, carbohydrates, and nucleic acids are degraded to their primary constituents [51]. Moreover, lysosomes are key actors of the autophagy machinery [52]. The link between early endosomes and lysosomes is not linear and there are several branches and loops, including those involving the Golgi apparatus and the endoplasmic reticulum [53,54]. Only a small fraction of ONs can escape to avoid degradation in the lysosomal environment, thus becoming available at the site of action [55]. In general, single-stranded ONs, being generally either small, uncharged, hydrophobic, or in combination, can be considered to be capable to productively enter cells and escape endosomes through a process called gymnosis [56], without the use of a delivery agent, and for which, however, there is the need of relatively high doses of ONs [43].

On the other hand, the vast majority of ONs are too large and charged to easily enter cells, thus requiring either delivery agents, chemical modifications, or both. These approaches are focused to accelerate the rate of cellular uptake and intracellular trafficking, as well as to facilitate endosomal escape, and can be subdivided into approaches for the direct conjugation to carriers and approaches for the loading in the nanoparticle carriers [41,55,57]. To be efficient, delivery systems must overcome the limits observed by the systemic administration of free ONs, ensuring the protection from premature degradation, the delivery to the target tissue, the cellular uptake and also the escape from the lysosomal environment [58]. With these aims, viral and non-viral vectors were produced. Viral carriers are widely used because of their high tolerability, standardized transfection procedures, great high and stable transfection efficiency, and high stability [59,60]. Other major features of viral vectors include their ability to infect specific cells through a process known as “tropism” and to transfer ONs in both dividing and non-dividing cells [60]. However, the use of viral vectors is limited by their difficult large-scale production, high production costs and a certain degree of immunogenicity and potential carcinogenicity [59,60]. Therefore, efforts were concentrated on the development of specific, safe and easy-to-fabricate non-viral vectors. Among them, remarkable carriers for the delivery of ONs are represented by lipidic, polymeric, and inorganic nanoparticles (NPs), also combined with each other (hybrids).

### 3.1. Lipid Nanoparticles

Lipid nanoparticles (LNPs) gained much attention in the 1980s when Felgner and colleagues demonstrated that the association of plasmid DNA (pDNA) with cationic lipids such as 1,2-di-*O*-octade-cenyl-3-trimethylammonium propane (DOTMA) and dioleyl phosphatidylethanolamine (DOPE) induced the in vitro and, lately, the in vivo transfection of cells [61]. All the subsequent studies on LNPs culminated in 2018 with the US Food and Drug Administration (FDA) approval of Patisiran (ONPATTRO) as a carrier for siRNA delivery into cells for the treatment of hereditary transthyretin (hATTR) amyloidosis [62]. Due to successful results in preclinical and clinical studies, LNPs are currently the only structures approved by the FDA as drug delivery carriers. As a matter of fact, by now, there are eight FDA- and European Medicine Association (EMA)-approved LNPs, but none of them are used for the delivery of ONs. In the way, a variety of RNA-LNPs have recently entered clinical trials and 7 out of 13 have been proposed for tumor treatment [63]. LNPs are classified into 5 different categories on the basis of their synthesis method and physicochemical properties: liposomes, niosomes, transfersomes, solid lipid nanoparticles (SLNPs), and nanostructured lipid carriers (NLCs) [63]. Liposomes represent the traditional type of LNPs and are composed of phospholipids and cholesterol. They constitute the most part—5 out of 8—of FDA/EMA-approved LNPs. Together with liposomes, niosomes and transfersomes are also lipophilic bi/multi-layer NPs. This particular structure allows the encapsulation of hydrophobic and hydrophilic drugs, which are preferentially incorporated in the external bilayer and the aqueous inner core, respectively. Despite their advantages such as the protection of the cargo, the sustainable drug release, and improved bioavailability, they suffer from some limitations such as toxicity, low loading ability, and their limited stability. SLNPs lack an aqueous core improving the loading efficiency of hydrophobic drugs in respect to liposomes. Furthermore, their rigid core increases their stability. NLCs have an imperfect crystal interior formed by a mixture of solid and liquid lipids which dramatically increases drug loading, also inhibiting drug release during storage [63].

In their modern definition, LNPs have a dimension of less than 1 µm and contain phospholipids, lipid anchored polyethylene glycol (PEG), cholesterol, and cationic/ionizable amino lipids (Figure 6A).

PEG and cholesterol are known to mitigate macrophages’ engulfment thus decelerating LNPs clearance, even though at the same time they can activate the immune system mostly through complement activation [61,64,65]. Cationic lipids have a net positive charge, which allows the efficient incorporation of negatively charged ONs and increases cellular uptake and endosomal escape. However, cationic lipids are susceptible to rapid clearance mediated by the reticuloendothelial system (RES) and their high positive charge is related to increased toxicity. To avoid these limitations, efforts were concentrated on the development of neutral lipids but with limited success [66]. The advent of ionizable amino lipids represents the turning point in the field.

Cationic/ionizable amino lipids are negatively charged when protonated at a pH below their pKa, ensuring high encapsulation efficiency of negatively charged ONs. Moreover, they are neutrally charged in physiological environments (pH < LNPs pKa), decreasing cytotoxicity and improving circulation. Finally, they are positively charged in an acidic environment (i.e., endosomes) leading to the association to anionic endosomal lipids and consequently the release of the payload in the cytosol [61,64]. Such formulations were demonstrated to be efficient in the in vitro delivery of siRNAs [67] and in vivo release of miRNAs [68,69,70], alone or in combination with other therapeutics, and ASOs [71].

LNPs have been studied for more than two decades as vehicles for ONs delivery; however, there are still several drawbacks limiting their use in humans. One example is represented by MRX34, a liposomal formulation of miRNA-34a mimic, which in 2013 entered the first in-human Phase I study for the treatment of hepatocellular carcinoma (HCC). Despite encouraging results obtained in an orthotopic mouse model of HCC [72] and in HCC-affected patients [73], a trial started in 2014 was prematurely closed due to serious immune-related adverse effects [74]. It is now known that LNPs can themselves induce toxicity and elicit an immune response [65,75]. To decrease these side effects, novel ionizable lipids, such as autotaxin (ATX), with a safer profile and higher miRNA deliver efficiency were developed [63].

### 3.2. Polymeric Nanoparticles

Polymeric NPs are less clinically advanced than lipid-based carriers but their integrity, their stability, and the possibility of being largely engineered make them interesting carriers for ON delivery (Figure 6B). Polymeric NPs are composed of synthetic or natural polymers. Among synthetic polymers, polyethyleneimine (PEI) was the most extensively studied being an effective gene transfection agent. The high number of amine groups present in its structure confers to PEI a positive charge, allowing strong affinity for ONs and facilitating interaction with the cell membrane. Even so, the high density of positive charges is related to in vivo toxicity and the activation of the immune system [76,77]. To avoid these issues, copolymeric structures, obtained through the association of PEI with a negatively charged biodegradable and biocompatible polymer such as poly(3-hydroxybutyrate) (PHB) [78], poly-(d,l-lactide-co-glycolide) (PLGA) [79], or also PEG, demonstrated an improved biocompatibility and efficient miRNA delivery. Furthermore, the employment of polymers such as poly(2-dimethylaminoethyl acrylate) (PDMAEA), which are positively charged at physiological pH and degrades into negatively charged non-toxic polymers in water, may offer a valid alternative for siRNA protection against enzymatic degradation and their efficient delivery into cells [80,81].

Among synthetic polymer-based NPs, dendrimers were extensively studied as ON delivery systems. In detail, dendrimers are chemically and physically stable, highly branched, three-dimensional molecules, 1–10 nm in size [60]. What makes them excellent candidates for ON delivery is their ability to carry and protect large amounts of ONs from degradation; moreover, the protonation of the abundant tertiary amines present in their structure facilitates the release of the cargo through the disruption of the endosomal membrane [59]. The most employed dendrimers are derived from poly(amidoamine) (PAMAM), a synthetic hydrophilic, biocompatible, and non-immunogenic polymer [59,60]. Promising results on the fifth generation of PAMAM dendrimers (PAMAM G5) were obtained. PAMAM G5 were in fact demonstrated to deliver in vitro miRNAs in glioma cells [82], and to release siRNAs in HCC cells in vivo [83].

Natural polymers offer an alternative to synthetic ones. Among them, chitosan has gained great interest because of its high positive charge associated with biocompatibility, low immunogenicity, and low cytotoxicity [76]. For an effective transfection, the endosomal escape is fundamental. Chitosan can exploit its buffer capacity in a pH interval between 5 and 7, promoting the rupture of endosomes after 72 h with the consequent release of the cargo in the cytoplasm [84]. The transfection efficiency of chitosan NPs also depends on its degree of deacetylation (DD) and molecular weight (MW) [76]. The DD determines the positive charge and the solubility of chitosan, the binding capacity, and the transfection efficiency. A DD higher than 80% is required for the release of siRNA [85]. Likewise, low MW polymer was reported to easily release siRNA in the cytoplasm and to favor endosomal escape with higher efficacy than the high MW counterpart [86]. Chitosan NPs were demonstrated to efficiently deliver miRNA when totally constituting the structure of vehicles [87] or as a copolymer with PLGA [88]. However, the transfection efficiency of chitosan is relatively low due to in vivo instability and insufficient cellular release.

### 3.3. Inorganic Nanoparticles

In the last decade, inorganic NPs (INPs) have been extensively studied because of their tunable size and surface properties, chemical and thermal stability, and low toxicity [62,89]. Of the INPs, gold NPs (AuNPs) are the most studied because they can be produced with low size dispersity and they can be functionalized by the creation of multifunctional monolayers to modulate cytotoxicity, biodistribution, and excretion [89]. ONs can interact with AuNPs through covalent or electrostatic interactions. Covalent attachment can be made just when this modification does not inhibit the biological activity of the cargo. Noncovalent bound ONs can be achieved by functionalizing AuNPs with amino acids (e.g., lysine dendron-coated AuNPs) or producing mixed-monolayer-protected or layer-by-layer-fabricated AuNPs such as siRNA/PEI-AuNPs or PEI/siRNA/PEI/AuNPs [89]. AuNPs were also demonstrated to efficiently enhance stability of decoys ensuring their therapeutic effect against head and neck cancer (HNC) [90]. Regarding miRNAs, even though AuNPs were demonstrated to release these molecules per se in vitro in multiple myeloma (MM) cells [91], typically INPs suffer from low transfection efficiency related to the lack of condensation ability with ONs [92] (Figure 6C).

### 3.4. Exosomes

Exosomes are nanoscale extracellular vesicles (EVs) first described 25 years ago. They are natural carriers of mRNA, miRNA, and siRNA and are composed of a lipid bilayer derived from the inward budding of late endosomes of cells, from which they are released, and an inner aqueous core (Figure 6D). Notwithstanding they share similarities with the structure of liposomes, exosomes showed lower toxicity and immunogenicity, and also higher target specificity [93].

An important recent progression in the field is represented by the bioengineering of exosomes which can be loaded with specific therapeutic drugs to improve their biodistribution and safety profile [94]. Engineered exosomes were reported to specifically deliver siRNA in oral squamous cell carcinoma cells [95], pancreatic ductal adenocarcinoma cells [96], postoperative breast cancer-derived lung metastasis [97], human pancreatic cancer cells [98], gastric cancer cells [96], and lung cancer cells [99] resulting in the significant decrease of tumor growth in vitro and in vivo. Furthermore, miRNA were reported to be efficiently released by exosomes in breast cancer [100,101] and colon cancer cells [102].

### 3.5. Hybrids

All the previously described vehicles suffer from some limitations. These issues can be overcome with the production of advanced next-generation delivery systems based on the combination of different components taking advantage of the physical properties of both structures (Figure 6E). Hybrids based on PEI are the most frequently proposed as PEI has been demonstrated to be the most effective polymer in gene transfection. PAMAM, AuNPs, and LNPs were combined with PEI resulting in improved transfection efficiency [92]. However, the use of PEI is not always the best method to ameliorate such aspects; in fact, the high binding affinity between PEI linked to AuNPs and siRNA was demonstrated to limit the cytoplasm release of the silencing ON. Chitosan represents one alternative to this approach; in fact, modification of the surface of AuNPs with this polymer was demonstrated to efficiently deliver siRNA [103].

Furthermore, the combination of LNPs with AuNPs (AuroLNPs) showed promising results. Notably, AuroLNPs did not cause complement activation, evading recognition by the mononuclear phagocyte system (MPS) due to the neutral charge of the platform; moreover, AuroLNPs were demonstrated to accumulate in the tumor mass and to release siRNA in vivo significantly inhibiting the tumor growth in two human ovarian cancer models [104].

## 4. Zebrafish (ZF) as an Animal Model to Mimic Human Cancer

There are several approaches to generate human cancer in ZF (*Danio rerio*), such as the development of mutant and then transgenic fish lines, or transplantation of tumor cells [105,106]. Transgenic lines can be induced through chemical mutagenesis, irradiation mutagenesis, transposon-based, or viral vector insertional mutagenesis. However, until now, researchers forced the development of several cancer types by directly adding carcinogens to the water or acting into one-cell-stage ZF embryos, microinjecting exogenous DNA [107]. Interestingly, when exposed to carcinogens, ZF develops tumors in virtually all organs with similar histology to human tumors [108].

In regard to xenotransplantation, ZF models nowadays represent a well-known option for implementing strategies of personalized medicine, together with other models of patient-derived xenografts or patient-derived organoids [109,110]. The efficiency and the growth of human cancer cells in ZF was firstly reported in 2005 [111] when authors evidenced many similarities in the behavior of tumors developed in mammalian models, putting the basis for further employment of ZF into this field. When injecting foreign cells into an organism, the state of the immune system is one of the first things that must be considered. ZF’s adaptive immune system does not reach maturity until 4-weeks post-fertilization, when immature T and B cells reach the thymus [112]. This lack of immune defense in ZF embryos allows the establishment of xenografts without the need of additional immunosuppression agents, facilitating tumor cells growth and reducing the variability. Furthermore, the times in which the animals are manipulated were significantly reduced [112,113]. Xenografts can be induced in adult ZF, but just after immune system ablation to avoid engraftment rejection. Methods applied to achieve immunosuppression in this context are similar to mouse model approaches [107]. Another strength of embryo ZF is the higher rapidity of cancer development if compared to adults; in fact, tumor formation was evidenced 2 days after the injection. Consequently, embryos could be employed in projects that demand rapidity, such as imaging cancer progression or screening processes. By contrast, adults offer a more realistic in vivo model, as all of their organs and immune systems are completely developed, but cancer establishment requires from 10–14 days to 1 month [114].

The development of xenograft tumor models was optimized by Haldi et al. who evaluated the impact of the site of injection, age of transplant recipients, the amount of injected cells, and post-injection ZF care [115]. In particular, depending on the developmental stage of the ZF, the microinjection site of the tumor cells can vary. Numerous sites of injection, including the hindbrain ventricle [115] and intravenous routes [116,117], were tested, but the yolk sac [115,118,119] in 2-day-old embryos was demonstrated to be an ideal approach. This is due to the yolk sac providing a large site to house transplanted cells and facilitates manual transplantation in comparison with other smaller regions, such as the duct of Cuvier, caudal vein, or heart [107]. Regarding larvae’s care after xenotransplantation, a temperature of 32–36 °C was demonstrated to be optimal [115,120,121,122] because in this range both natural embryonic development of the ZF transplant avatar and satisfactory human cell survival and proliferation are ensured [120,121].

Xenotransplant in *Danio rerio* is also supported by the transparency of the embryos from fertilization to when pigmentation is initiated (at approximately 30–72 h post-fertilization (hpf)) and the tissues become denser [123]. This process can be easily inhibited by treating embryos with *N*-Phenylthiourea (PTU), a molecule which suppresses the production of melanin, allowing the direct imaging of animal development, organogenesis, and cancer progression, also enabling the tracking of transplanted cells—previously transfected with GFP or pretreated with fluorescent membrane stains—or fluorescent NPs [113]. Moreover, this interesting feature allows the live imaging of injected NPs in the bloodstream through fluorescence microscopy. Currently, the study of interaction between foreign structures—xeno-transplanted cells or NPs—and immune or blood cells is allowed by in vitro tests, such as synthetic microvascular network developed by Vu et al. [124]; the transparency of ZF larvae represents an important feature to translate this study into a more realistic system. In fact, the bloodstream of ZF is easily visualized in live imaging, permitting this kind of study. Through these methods, it is common and easy to successfully differentiate injected from non-injected embryos, although it is still challenging to obtain a reproducible volume and therefore number of cell or NPs administration [125]. However, there are particular characteristics of ZF that may be problematic. The most obvious shortcoming of ZF is that it is not a mammal and lacks a placenta, but also breast and lungs [121]. This characteristic impedes an entire category of experiments that should be performed in a different animal model, such as mice and rats. In addition, there is a difference in the incubation temperatures, while injected human cancer cells should be maintained at 37 °C, the optimal temperature for ZF is 28 °C. Thus, a compromise incubation temperature should be used, but possible metabolic changes could occur and should be considered [121,126].

Therefore, each methodology has several advantages and disadvantages, and experimentation should be planned to better address the aim of the study considering the ZF developmental stage, each one presenting some different benefits.

### ZF for In Vivo Characterization of New Drugs

ZF embryo is rapidly becoming an attractive and prominent in vivo vertebrate model for the screening of new drugs, nano-bio materials, and NPs, in regard to their biodistribution, toxicity [84,127,128], and therapeutic efficacy. ZF offers a variety of advantages as a desirable model for in vivo high-throughput drug screening. This is achieved through microscopic analysis platforms taking advantage of ZF translucency and its ability to be maintained in 96-well plates. The optical transparency of embryos not only offers exciting research opportunities, allowing visualization of the biodistribution of compounds at high resolution [128], but also permits the evaluation of other parameters, including a lower hatching rate, mortality, and malformations [127].

ZF represents a versatile animal model due to the possibility of testing new therapeutic strategies in different ways. Toxicity studies can be carried out by the dissolution of drugs in water or their direct injection into the egg, the embryo, or the adult fish [107]. In the first case, eggs and fishes are immerse in the solution containing the compound(s) and the exposure route (i.e., chorion and epidermis; epidermis; epidermis and intestine via ingestion) influences the uptake and biodistribution of particles in ZF embryos. The predominant route still remains the oral one, since uptake over an internal mucosal membrane occurs faster than over an intact dermal epithelial membrane. The main physicochemical factor accounting for uptake, internalization, and further distribution is the size of particles [129]. Regarding injection into embryos, these could be performed in the yolk or in the duct of Cuvier, in order to obtain a localized or diffused distribution, respectively. Instead, in adult ZF, there are additional exploitable sites of injection, such as the retroorbital or intra peritoneal routes. Generally, small drugs and NPs are mostly dissolved in water, while larger structures can be intravenously micro-injected, and, if previously labeled with a fluorescent dye, directly observed at the whole animal level [130] (Figure 7).

MPS represents an important player in the distribution of NPs in the body. In fact, the uptake by macrophages is generally considered to decrease the chances of NP accumulation in tumor sites and it is therefore considered a deleterious factor in terms of efficiency [131,132]. Early embryonic macrophages are already present in ZF embryos at 30 hpf and additional cells, which define the mammalian MPS, can be found in adult ZF. Another important cell type that affects NP distribution is represented by endothelial cells which bind and take up NPs [131]. On these bases, the wide availability of transgenic ZF, in comparison to mouse models, allows the assessment of NP interactions with selectively fluorescent cells such as macrophages (e.g., Tg(mpeg:mcherry)) [133], endothelial cells (e.g., Tg(Fli1a:EGFP)y1) [134], and neutrophils (e.g., Tg(mpx: GFP)i114) [135]. Another important characteristic to take into account before starting an experiment with an animal model is the temperature. In fact, as mentioned above, ZFs are routinely maintained at 28 °C, which differs by 10 degrees from that of the mouse (38 °C) [136] and 9 °C from that of the human body (37 °C), the latter being the ideal temperature for tumor cell proliferation [121]. Even if, the variation of the temperature of around ~10 °C did not show differences in the internalization behavior, at the moment there is still a paucity of information in this field, and more studies are needed to fully investigate cell membrane composition and behaviors at ZFs internal temperature.

In recent years, authors take advantage of ZF models to study and characterize ONs in both cancer and non-cancer related disorders. To the best of our knowledge, there is still paucity of information on this field. With regards to toxicity, some siRNAs injected into ZF embryos were demonstrated to induce morphological defects, abnormal development, and early death of animals [137], limiting the use of this model for these purposes. Regarding the efficacy of free ONs, a treatment based on ASOs was demonstrated to be efficient in restoring the exon skipping in a ZF model of retinitis pigmentosa [138], further suggesting the potential application of this model. Typically, ONs are not administered after dissolution in water, but they are fundamentally micro-injected in ZFs as free molecule or loaded inside NPs. In fact, ZF can be used for the characterization of ON-loaded NPs, which are known to change biodistribution of free molecules, consequently increasing their therapeutic effect and, in the meantime, decreasing toxicity. In fact, siRNA-loaded NPs were characterized in a ZF brain cancer model, demonstrating their efficacy through the decrease of cancer growth [139] and creating the basis for further studies.

## 5. Conclusions and Perspectives

ONs are rapidly growing as new therapeutic agents for the treatment of different forms of cancers; their versatility and the diverse chemical characteristics guarantee the possibility of targeting nucleic acids and proteins involved in either tumor development, progression, or both. Several issues have still to be solved for their clinical use but the development of specific delivery systems based on nanostructures allows overcoming most of them, showing a selective action of the ON in the tumor microenvironment. Moreover, the possibility to exploit new in vitro and in vivo models permits a rapid characterization of the new therapeutic approaches based on ONs; in this context ZF is obtaining popularity because of its capacity to provide strong results in a short period of time, in a system less complex than mice and particularly useful for the screening of new compounds or delivery systems (Figure 8).

Recent breakthroughs in vaccines based on mRNA exploit LNPs to guarantee nucleic acid protection but also to increase the delivery of mRNA vaccines in vivo [66]. Clinical studies clearly evidenced the potential results deriving from this approach, boosting research activities and clinical trials aiming to develop new therapeutic strategies, not only for the treatment of infectious diseases but also for other clinical manifestations, including cancer [63]. Several issues remain to be addressed in order to move to a prolonged or chronic treatment, linked, for examples, to the interaction of the nanostructures with the immune system of the patients; the formation of a protein corona opsonizing nanoparticles induces a rapid elimination of the delivery systems. The capacity to reduce this interaction and the possibility to better exploit efficient targeting agents remain important factors that have to be taken in consideration during the development of nanoparticle-based oligonucleotide delivery in cancer.

Taking together these aspects, NP-based delivery of ONs in the tumor microenvironment, with their characterization in ZF models, represents a future weapon for the treatment of several clinical manifestations; the development of effective delivery systems remains a pivotal issue and the ZF model contribute to their characterization, demonstrating to be an ideal approach for the screening of new therapeutics for the treatment of cancer.

## Figures and Tables

**Figure 1 pharmaceutics-13-01106-f001:**
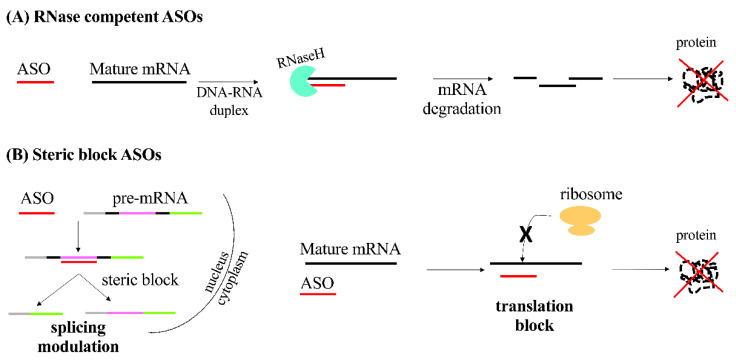
Therapeutic ASOs. Schematic representation of the mechanism of action of antisense oligonucleotides (ASOs) (**A**) RNase competent and (**B**) steric block. mRNA: messenger RNA.

**Figure 2 pharmaceutics-13-01106-f002:**
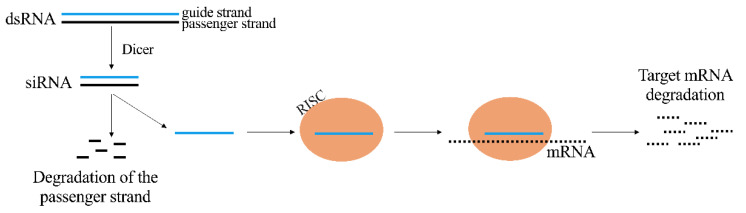
Mechanism of action of therapeutic siRNA. dsRNA: double strand RNA; siRNA: short interference RNA; RISC: RNA-induced silencing complex; mRNA: messenger RNA.

**Figure 3 pharmaceutics-13-01106-f003:**
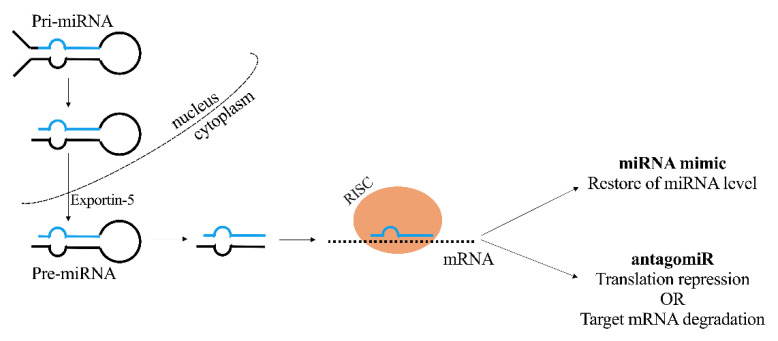
Mechanism of action of therapeutic miRNA. miRNA: microRNA; RISC: RNA-induced silencing complex; antagomiR: anti-miRNA; mRNA: messenger RNA.

**Figure 4 pharmaceutics-13-01106-f004:**
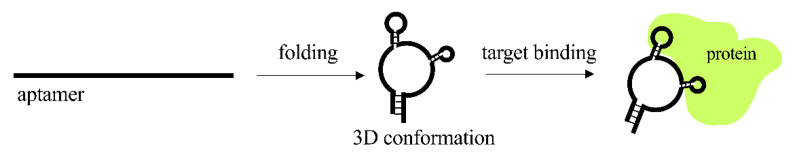
Mechanism of action of aptamers.

**Figure 5 pharmaceutics-13-01106-f005:**
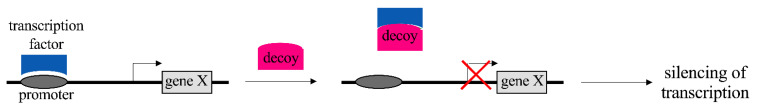
Representation of the mechanism of action of decoys.

**Figure 6 pharmaceutics-13-01106-f006:**
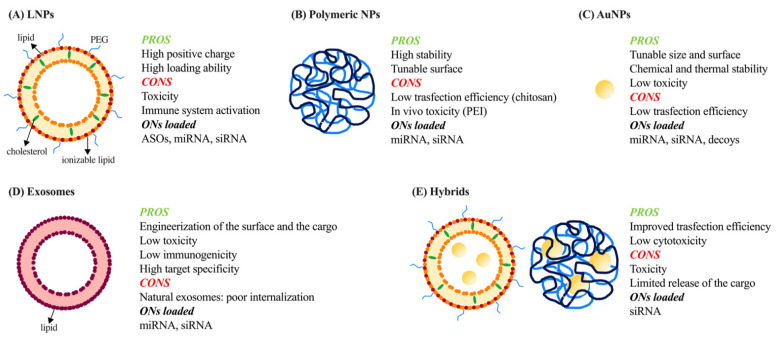
Schematic representation of different types of NPs for ONs delivery. Pros and cons of different types of nanoparticles (NPs) used for oligonucleotides (ONs) delivery: (**A**) lipid nanoparticles (LNPs), (**B**) polymeric NPs, (**C**) gold NPs (AuNPs), (**D**) exosomes and (**E**) hybrids. PEG: polyethylene glycol; ASOs: antisense oligonucleotides; miRNAs: microRNAs; siRNAs: small interfering RNAs; PEI: polyethyleneimine.

**Figure 7 pharmaceutics-13-01106-f007:**
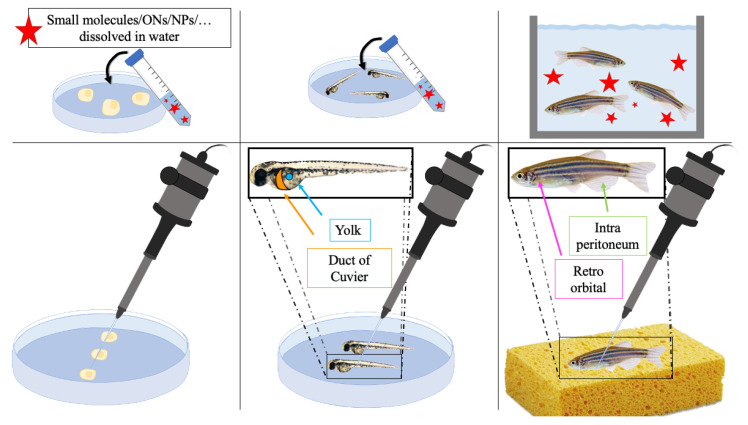
Schematic representation of different strategies to administer therapeutic compounds. ONs: oligonucleotides; NPs: nanoparticles.

**Figure 8 pharmaceutics-13-01106-f008:**
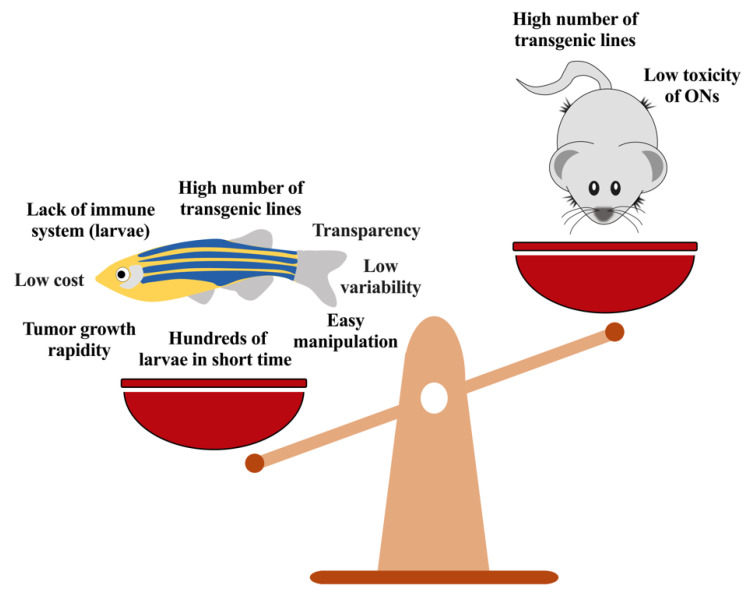
Zebrafish vs. mouse models. ONs: oligonucleotides.

**Table 1 pharmaceutics-13-01106-t001:** Therapeutic oligonucleotides (ONs). ASO: antisense oligonucleotides; DNA: deoxyribonucleic acid; RNA: ribonucleic acid; ssDNA: single strand DNA; dsDNA: double strand DNA; ssRNA: single strand RNA; dsRNA: double strand RNA; siRNA: small interfering RNA; RISC: RNA induced silencing complex; mRNA: messenger RNA; miRNA: microRNA; antagomiR: anti-miRNA.

ONs	Nucleic Acid Composition	Lenght (Nucleotides)	Mechanisms of Action	Effects
ASO	ssDNA or ssRNA	18–30	Formation of RNA-DNA duplex recognized by RNASEH1	Silencing of gene expression after RNA degradation
			Steric block	Modulation of alternative splicing or translation block
siRNA	dsRNA	21	RISC is guided to the complementary target transcripts	Degradation of the target mRNA, gene silencing
miRNA mimics	dsRNA	~20	Increased level of a specific miRNA	Restore of miRNAs level
antagomiR	ssRNA	~20	Direct binding to RISC	Translation repression or miRNA degradation
Aptamers	ssDNA or ssRNA	20–100	Mimic of the promoter sequence	Block of protein translation
Decoys	dsDNA or dsRNA	15–20	Mimic binding site of transcription factors	Transcription silencing

## Data Availability

Not applicable.

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
