# Peer review of "Nanoparticles-Based Oligonucleotides Delivery in Cancer: Role of Zebrafish as Animal Model"

_pharmaceutics, 2021, doi:10.3390/pharmaceutics13081106_

Round 1

Reviewer 1 Report

In this short but comprehensive review, Macor and co-workers summarize the oligonucleotides therapeutics, their delivery systems, and the zebrafish model. I found this review great important to the field and interesting to read. Oligonucleotides therapeutics hold great promise as future medicine, while the nanoparticle-based delivery vesicles are the key to translate these oligonucleotides therapeutics into the clinics. The authors also highlight the zebrafish as a suitable animal model exhibiting many advantages such as low cost, rapid tumour growth, easy manipulation, transparency, lack of immune system, a high number of transgenic lines. Overall, the review is well-written and should be published in Pharmaceutics after revisions. I have only a few comments to improve the manuscript, as noted below.

  1. The authors may consider adding another Figure summarizing all the oligonucleotides therapeutics discussed in this review (similar to Figure 6 in which the authors nicely summarize the delivery systems). If possible, please also highlight their similarity and difference in structure and mechanism of action.
  2. Section 3.1 lipid nanoparticles is quite short. So the authors may refer the readers to two recent reviews that have more comprehensive information and discussion about lipid nanoparticles: https://doi.org/10.3390/vaccines9040359 and https://doi.org/10.1016/j.actbio.2021.06.023
  3. The sentence “Due to successful results in preclinical and clinical studies, liposomes are currently the only nanoparticles approved by FDA as drug delivery carriers” should be revised. As the authors pointed out, Onpattro is also approved by FDA as siRNA carrier, and the Onpattro is a solid lipid nanoparticle (not a liposome with a hollow water core). It is also noted that lipid nanoparticles are also used in mRNA vaccines (Pfizer/BioNtech and Moderna).
  4. In section 3.2 polymeric nanoparticles, the sentence “To avoid these issues, copolymeric structures obtained through the association of PEI with a negatively charged biodegradable and biocompatible polymer such as poly(3-hydroxybutyrate) (PHB) [57], poly-(D,L-lactide-co-glycolide) (PLGA) or also PEG demonstrated an improved biocompatibility and efficient miRNAs delivery” should also discuss and cite two more references using PDMAEA to avoid these issues of positive charges: https://doi.org/10.1038/ncomms2905 and https://doi.org/10.1021/bm2007423
  5. In the section 4.1 zebrafish for in vivo characterization of new drugs, the authors should add another advantage of optical transparency of embryos is to characterize nanoparticles under flow conditions and cite a relevant paper (https://doi.org/10.1002/smll.202002861) for more details on the importance of characterization under flow conditions.
  6. There is no doubt that zebrafish is a good animal model for testing nanoparticles delivery systems. That said, the review would be more balance to also mention some drawbacks that other animal models should be considered when needed.
  7. Last but not least, it is great to hear more from the authors about the challenges in the field and future research directions to translate oligonucleotides therapeutics into the clinics.

Author Response

Response to Reviewer 1 Comments

We thank the reviewer for finding our review “great important to the field and interesting to read” and we are grateful for the comments that help to improve and complete the message of our work.

Point 1: The authors may consider adding another Figure summarizing all the oligonucleotides therapeutics discussed in this review (similar to Figure 6 in which the authors nicely summarize the delivery systems). If possible, please also highlight their similarity and difference in structure and mechanism of action.

Response 1: We summarized these information in a new table (Table 1) for an immediate visualization.

Point 2: Section 3.1 lipid nanoparticles is quite short. So the authors may refer the readers to two recent reviews that have more comprehensive information and discussion about lipid nanoparticles: https://doi.org/10.3390/vaccines9040359 and https://doi.org/10.1016/j.actbio.2021.06.023

Response 2: This section has been modified including new information and new citations (including scientific articles proposed by the reviewer).

Point 3: The sentence “Due to successful results in preclinical and clinical studies, liposomes are currently the only nanoparticles approved by FDA as drug delivery carriers” should be revised. As the authors pointed out, Onpattro is also approved by FDA as siRNA carrier, and the Onpattro is a solid lipid nanoparticle (not a liposome with a hollow water core). It is also noted that lipid nanoparticles are also used in mRNA vaccines (Pfizer/BioNtech and Moderna).

Response 3: Thanks to have pointed out the error in the text, that has been modified.
We have focused our manuscript on cancer and initially we were not sure to include also data on mRNA-based vaccines. Taking in consideration that these new approaches will speed up the development of new therapeutics also for cancer treatment, we have included some information in the final paragraph.

Point 4: In section 3.2 polymeric nanoparticles, the sentence “To avoid these issues, copolymeric structures obtained through the association of PEI with a negatively charged biodegradable and biocompatible polymer such as poly(3-hydroxybutyrate) (PHB) [57], poly- (D,L-lactide-co-glycolide) (PLGA) or also PEG demonstrated an improved biocompatibility and efficient miRNAs delivery” should also discuss and cite two more references using PDMAEA to avoid these issues of positive charges: https://doi.org/10.1038/ncomms2905 and https://doi.org/10.1021/bm2007423

Response 4: This section has been revised including also new citations.

Point 5: In the section 4.1 zebrafish for in vivo characterization of new drugs, the authors should add another advantage of optical transparency of embryos is to characterize nanoparticles under flow conditions and cite a relevant paper

(https://doi.org/10.1002/smll.202002861) for more details on the importance of characterization under flow conditions.

Response 5: This section has also been modified including this important advantage (we had not focused) and new citations.

Point 6: There is no doubt that zebrafish is a good animal model for testing nanoparticles delivery systems. That said, the review would be more balance to also mention some drawbacks that other animal models should be considered when needed.

Response 6: Thanks for this comment. The manuscript has been modified in sections 4 and 4.1, resulting more balanced and complete.

Point 7: Last but not least, it is great to hear more from the authors about the challenges in the field and future research directions to translate oligonucleotides therapeutics into the clinics.

Response 7: Thanks also for this comment. “Conclusion and perspective” include now some future studies we think have to be performed taking in consideration the massive elimination of nanostructures by cells of the immune system.

Reviewer 2 Report

“The chemical structure of oligonucleotide compounds could affect their capability to pass through 14 the plasma membrane.“

This is massively understating the problem. The limiting membranes of endocytic vesicles have evolved to compartmentalise large molecules and do this effectively. Consequently, getting ASOs and siRNA out of the endocytic system and into the cytosol is a problem that needs addressing.

Line 17. What about exosomes? These are nanosized particles !

Section 3 might talk about the COVID vaccines and exosomes ?

Section 4.

Discussing the differences in physiology specifically temperature and endocytosis between ZF and mammals might be helpful. These might cell membrane composition and behaviours at a different temp? Lipid composition? Endocytic fusion accuracy?  Increased decreased vesicle leakage?

Author Response

Response to Reviewer 2 Comments

We thank the reviewer for the comments that help to improve and complete the message of the manuscript.

Point 1: “The chemical structure of oligonucleotide compounds could affect their capability to pass through the plasma membrane.”. This is massively understating the problem. The limiting membranes of endocytic vesicles have evolved to compartmentalise large molecules and do this effectively. Consequently, getting ASOs and siRNA out of the endocytic system and into the cytosol is a problem that needs addressing.

Response 1: This is the starting point when we think about this manuscript. We modified section 3 providing more information on this aspect.

Point 2: Line 17. What about exosomes? These are nanosized particles

Response 2: We now included a paragraph (3.4) dedicated to exosomes. Thanks for have pointed out it.

Point 3: Section 3 might talk about the COVID vaccines and exosomes ?

Response 3: Section 3.4 is now dedicated to exosomes.. We have focused our manuscript on cancer and initially we were not sure to include also data on mRNA-based vaccines. Taking in consideration that these new approaches will speed up the development of new therapeutics also for cancer treatment, we have included some information in the final paragraph.

Point 4: Discussing the differences in physiology specifically temperature and endocytosis between ZF and mammals might be helpful. These might cell membrane composition and behaviours at a different temp? Lipid composition? Endocytic fusion accuracy? Increased decreased vesicle leakage?

Response 4: Section 4 has been revised including the information even if this important point has to be further addressed with specific studies.

Reviewer 3 Report

Overall this is an interesting paper that sheds light on the use of zebrafish in oligonucleotide delivery. 

However, it is poorly written with the English needing much revision.  

Line 70/71- rephrase

Line 104 – siRNA not siRNa

Line 113 – spelling of “synthesized” to be corrected

These were just a few of such corrections. Authors need to do a thorough review of the grammar. Apart from this, I have just one comment for improvement of the manuscript.

Section 2.4. Aptamers – I suggest a bit more detail into the mechanism of action be included in this section.

Author Response

Response to Reviewer 3 Comments

We thank the reviewer for the comment: “Overall this is an interesting paper that sheds light on the use of zebrafish in oligonucleotide delivery”. The manuscript has been sent for grammar revision, maintaing “American English” as indicated in the instruction.

Point 1: Section 2.4. Aptamers – I suggest a bit more detail into the mechanism of action be included in this section.

Response 1: Section 4 has been revised with the aim to include more details into the mechanism of action used by aptamers.
